# A Decision System for Routing Problems and Rescheduling Issues Using Unmanned Aerial Vehicles

**I-Ching Lin [1,2,*], Tsan-Hwan Lin [3] and Sheng-Hung Chang [4]**

1 Department of Transportation and Communication, National Cheng Kung University, Tainan City 701, Taiwan
2 Department of Management, Air Force Institute of Technology, Kaohsiung City 82047, Taiwan
3 Department of Logistics Management, National Kaohsiung University of Science and Technology, Kaohsiung City 807618, Taiwan; percy@nkust.edu.tw
4 Department of Industrial Engineering and Management, Ming Hsin University of Science and Technology, Hsinchu 30401, Taiwan; shchang@must.edu.tw
* Correspondence: yclin70626@gmail.com; Tel.: +886-6-5757575 (ext. 62421)

**Abstract:** In recent years, consumers have come to expect faster and better delivery services. Logistics companies, therefore, must implement innovative technologies or services in their logistics processes. It is critical to adopt unmanned aerial vehicles (UAV) in last mile delivery and urban logistics. The service provider applies the characteristics of UAVs to complete more requests, benefiting more revenue. However, it may not be a satisfactory solution, because the customers will be dissatisfied if the actual delivery time does not align with their expectations. This study constructs a revenue maximization model subject to time windows and customer satisfaction. Instead of addressing the traveling salesmen problem, this model takes new customer requests during the delivery process into account. We solved the problem using a genetic algorithm. The results show: (1) the model found an approximate and effective solution in the real-time delivery environment; (2) customer satisfaction is inversely proportional to the total delivery distance; (3) regarding the result of the sensitivity analysis of this study, investment in UAV has no influence on total profit and customer satisfaction. Moreover, the customer is a key factor in the logistics decision-making platform, not the provider's investment in UAVs.

**Keywords:** traveling salesmen problem; unmanned aerial vehicle; UAV; rescheduling

## 1. Introduction

Unmanned aerial vehicles (UAVs), an emerging technological product, have been applied in the last mile of delivery in cities to resolve the uneven quality of delivery personnel and slow delivery speed. Currently, topics related to unmanned vehicles are gradually getting the attention of researchers and the logistics industry. Many companies, e.g., Amazon (Seattle, WA, USA), DHL (Bonn, Germany), Google (Mountain View, CA, USA), United Parcel Service (UPS) (Atlanta, GA, USA), JD.com (Beijing, China), and SF Technology (Shenzhen, China), continue investing in delivery methods that take advantage of the utilization of UAVs. More than one hundred cities started running unmanned vehicle trial operations [1,2]. The thriving development of e-commerce has brought the logistics industry great profit and challenges. Consumers can easily retrieve logistic information through mobile phones and computers, increasing requests for prompt delivery. Being the essential value-added factor, the last mile of delivery has always been a difficult challenge. Due to the limitations of congested urban roads, as well as the physical strength, organization, and sense of direction of the delivery personnel, technological improvement is a critical direction in transformations for urban delivery. These factors/limitations influence the decision-making of urban transport performance [3]. Some modern technologies have been rolled out to solve the problem of urban transport. Particularly, introducing UAVs to

logistics will result in extensive changes and a positive influence on the environment in its current state. UAVs are aircraft without a human pilot. They can complete basic, special, or dangerous tasks, such as pesticide spraying, fire rescue, geological and archeological exploration, and logistic transportation, without being limited by reaching difficult terrains or places [4]. In the last three years, some literature has presented the application and limitations of UAVs in emergencies in the health care industry, especially during the COVID-19 outbreak [5]. It extends its' service in areas such as monitoring public gatherings and keeping the public informed, facilitating construction, and drone delivery [6]. However, due to restrictions related to city development, congested urban roads, inadequate infrastructure, and the route planning startup phase, UAV delivery services will lead to longer delivery time and higher costs [7]. If UAVs are joined to an existing supply chain, all processes concerning what is stocked, transported, and received at multiple locations will have to change to account for a new method of transport. Therefore, logistics systems proposed optimized routing that requires a new set of variables to determine the most efficient routing.

The time spent by vehicles traveling in specific areas needs to be optimized because of their limitations in sensory range and delivery capability. Route planning is one of the most challenging issues in UAVs research [1,2,8]. In general, we can solve this problem through a traveling salesman problem (TSP), which is a special type of vehicle routing problem (VRP). The difference between TSP and VRP is shown in Figure 1, in which the TSP is a single-route node-service-combination problem without the capacity limitation, and a VRP is a multiple-route node-service-combination problem with vehicle capacity limitation, such as cost controls, time windows, and resource limitations concerning the loading process at the depot, etc. Considering the characteristics of UAVs, and the combinations of the cities for the corresponding routing, TSP can help find the solution in searching for the shortest possible route that visits each city and returns to the origin city.

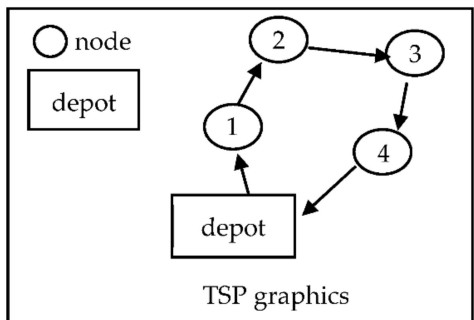 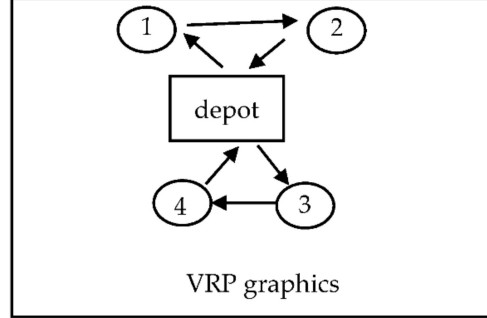

**Figure 1.** Difference between TSP and VRP.

Route planning was based on personal experiences in the early days [9]. Many studies in the literature have proven that these problems all belong to the class of NP-hard problems [10] and have proposed different algorithms to calculate solutions for the TSP or VRPs with the potential for a better solution [7,11,12]. Table 1 shows as classification of the methods of TSP and VRP.

**Table 1.** Classification of the methods of TSP and VRP.

| Category | Subcategory | Authors |
|---|---|---|
| | particle swarm optimization (PSO) | Khosiawan et al. [4] |
| heuristic algorithm | tabu search | Zhen et al. [8] |
| | genetic-sweep algorithm | Euchi and Sadok [13] |
| mixed-integer linear programming | | Ozkan and Atli [14] |
| swarm intelligence | | Schwarzrock et al. [11] |
| Game theory | | Gigante et al. [15] |

Table 2 shows the dimension of the decision maker considered reflected in TSP/VRP, including the operation time and organization. In addition, some additional factors are considered in the UAVs studies.

**Table 2.** The considered factors of TSP and VRP.

| Dimension | Factors | Authors |
|---|---|---|
| Operation time | Customer due date, customer service time, processing time, setup time, release date, ready time, and idle time | Chen et al. [16], Dewa et al. [17] |
| Organization | Leadership, development process of the system, and governance | Barmponuakis et al. [18], Eichleay et al. [10] |
| UAVs | Robust wireless communication, three-dimensional trajectory data, precise UAV control, weather conditions, air traffic control, fuel consumption, and service range, socio-technical | Khosiawan et al. [4], Thibbotuwawa et al. [9], Dingil et al. [3] |

The purpose of the traditional TSP is to minimize the total operating cost or travel time. In fact, with the development of e-commerce, it is even more important to meet customer delivery requests; the logistics service provider is paying more attention to customer satisfaction [19–21]. We called it a TSP, with time windows, noted as TSPTW. Past literature has included time windows and customer satisfaction in the route planning problem. Customer satisfaction is correlated to logistic time windows in the logistics industry; customers will be dissatisfied if the expected customer time window does not cover the customer's expected time, such as the actual delivered time, later or earlier than this [22]. At the same time, a penalty factor is also enforced when the customer which required the delivery time window is not satisfied; normally, the penalty factor, such as delivery cost or punishment, for exceeding the upper limit of the customer's expected time window is greater than not reaching the lower limit of the customer's expected time window. Nevertheless, some instabilities, or new requests coming in, might occur to route planning during implementation, and the initial plan might no longer be feasible. In this circumstance, the allocation of resources should be adjusted, or rearranged, due to the need for more resources [17,23–25].

Given that UAV studies are receiving increased attention from the logistics service providers, to allow for a faster logistics delivery speed and a higher resource application rate, this study will consider the delivery service process as a production schedule and expect rush requests during the production process. Under the circumstances of not exceeding the customer's expected time window and to guarantee customer satisfaction and the capacity limit of unmanned vehicles, new and timely logistics activities will be enforced to increase logistical efficiency and benefit while unmanned vehicles are processing the original logistical activity. Finally, the remainder of the paper is structured as follows: Section 2 explains the problem and detailed framework of the proposed scheduling system for the UAVs. Section 3 describes the key elements involved in the implementation of the heuristic algorithm and the proposed methodology. Section 4 discusses numerical experiments and the results of the implemented methodology. Section 5 concludes the findings of this research.

## 2. Problem Description of This Study

Along with the continuous development of e-commerce, consumers have a higher desire for logistics delivery speed. Under this premise, the logistics service provider applied the characteristics of unmanned vehicles to complete more requests and obtain more revenue. Logistics delivery speed efficiency will be enhanced to build the enterprise's competitive advantages if they can focus on effectively solving the route planning problems. Differing from the traditional delivery model, the logistics service provider provided a vehicle in order to allow the customer's contracted shipper (Manufacturer 1) to deliver the product to the receiver (Client 1). The logistics service provider will plan the best delivery route for the decision-making platform that is waiting to be delivered first, and start the delivery process. However, the platform built by the logistics service provider will receive a new delivery request (Client 2 has a new request to Manufacturer 2) in the UAV delivery process. The logistics service provider will determine whether or not to

take the new request during the UAV delivery process. Therefore, the platform must be provided with the solution for the logistics service provider in order to discuss whether or not it will impact the existing routing and consider delaying the delivery time for the customer. The business model of the logistics service provider is shown in Figure 2. This study has established an equivalent mathematical model that focuses on the new route planning problem of the logistics service provider, and further designs a heuristic algorithm to effectively resolve this kind of route planning problem.

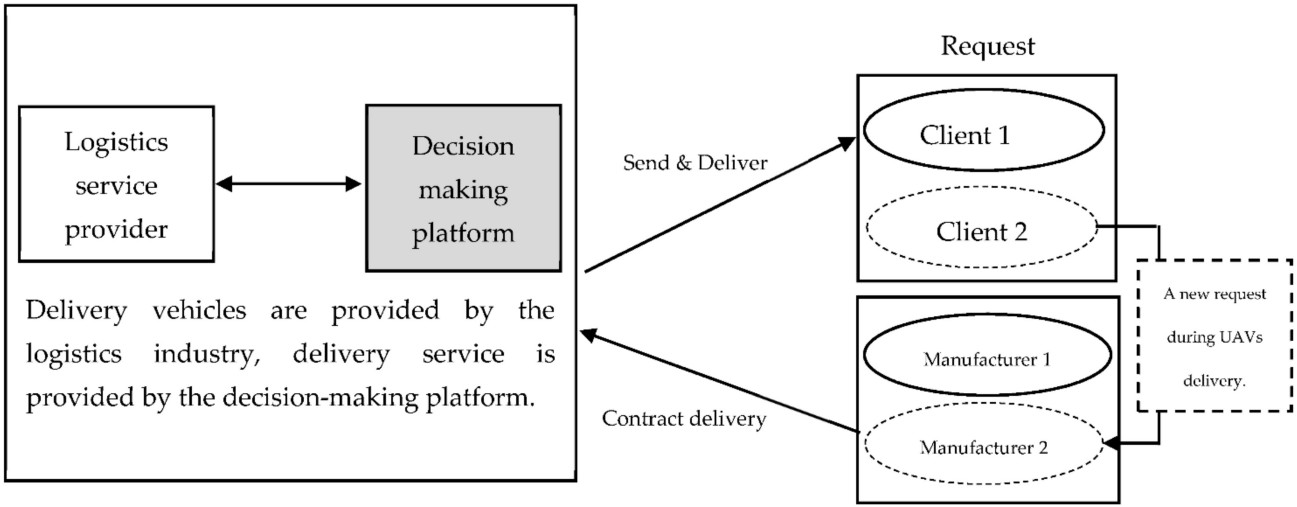

**Figure 2.** The business model of logistics industry delivery.

The greatest feature of this study is the decision-making platform of the logistics service provider. The platform combines the route planning problem model and production schedule model and provides the best routing solution to the logistics service provider during the delivery process. Traditionally, the process of receiving a request and scheduling delivery are considered two separate functions; this study applied the related concept models from past literature as references to establish the platform decision-making analysis procedure [22,26], shown in Figure 3, which is processed in two states: match and real-time. In the match state of the decision-making platform, the request is serviced by unmanned vehicles in the logistics center; the solution is provided that obtains the minimization of delivery cost combined with the known conditions of the number of requests, delivery conditions, vehicle volume, delivery region, and the customer's expected time window. The traditional routing scheduling is realized to provide the solution to the logistics service providers. However, new requests might be received during UAVs delivery, and the routing scheduled in the match state may no longer be the best solution due to these changes. In the real-time state, the platform considers customer service time gaps, route gaps of the serving vehicle, and the logistics cost of the service provider to determine whether or not the request should be taken and delivered in real-time. If the solution of all the considered factors by the logistics service provider cannot be satisfied, the new request is not serviced in real-time. It will be processed after the new request is delivered to the logistics center to find the best route that optimizes the logistics service provider's revenue. This structure will allow the logistics service provider to make the best decision in real-time, while increasing the revenue and maintaining customer satisfaction.

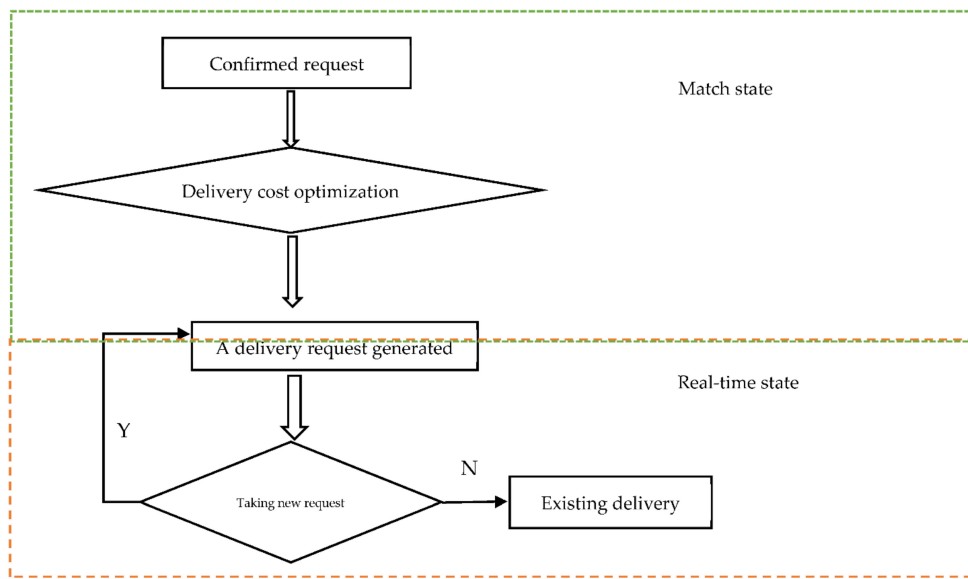

**Figure 3.** The analysis process of the decision-making platform.

## 3. Mathematical Model of This Study

In this section, a mathematical model is constructed in order to solve the problem in the study. We defined the symbols, assumed the hypothesis, and developed a heuristic algorithm to solve the problem.

### 3.1. Symbols

Table 3 shows the symbol definitions of the developed model.

**Table 3.** The variables used in the developed model.

| Decision Variable | Description |
| --- | --- |
| $x_{ijk}$ | The binary variable of 0 or 1<br>0: other situations, 1: unmanned vehicle $i$ travels from $j$ to $jk$ |
| **Parameter definition** | |
| $W_1$ | The weight of logistics service provider's profit |
| $W_2$ | The weight of the total distance |
| $W_3$ | The weight of penalty |
| $i$ | Unmanned vehicles with different models and capacities, $i = 1 \ldots , D$; we assumed that it was only one in this study |
| $j$ | Customer in match state, $j = 1 \ldots , C$ |
| $jk$ | Customer in the real-time state, $jk = 1 \ldots , C$ |
| $M$ | Customer number (a big constant); we assumed that it was 999 in this study |
| $p$ | Delivery price (Fees charged by logistics service providers to customers) |
| $Q_i$ | The capacity of unmanned vehicles $i$ |
| $FC_i$ | Fixed travel cost of unmanned vehicles $i$ |
| $VC_i$ | Variable travel cost of unmanned vehicles $i$ |
| $T_{i,0}$ | The initial time of unmanned vehicle $i$; we assumed that it was 0 in this study |
| $T_{i,max}$ | Maximum travel time of unmanned vehicle $i$ |
| $q_j$ | Customer $j$'s requirement |
| $q_{jk}$ | Customer's rush requirements in real-time |
| $l_i$ | The load of the unmanned vehicle is inversely proportional to the flight time |
| $d_{jk}$ | Unit distance from the customer in real-time state |
| $v_i$ | The unit distance required speed of unmanned vehicle $i$ |
| $c_{j,b}$ | The early fees of an unmanned vehicle arriving before the start of customer $j$'s time window |
| $c_{j,e}$ | The delay fees of an unmanned vehicle arriving after the end of customer $j$'s time window |
| $c_j$ | The fees of the unmanned vehicle |
| $T_{(i,d)}$ | Total operating time of unmanned vehicle $i$ |
| $T_{(j,a)}, T_{(k,a)}$ | The time arrived the customer in the real-time state of the unmanned vehicle ($a$-arrive) |
| $T_{(k,b)}, T_{(j,b)}$ | Time window lower limit of customer in real-time state |
| $T_{(k,e)}, T_{(j,e)}$ | Time window upper limit of the customer in the real-time state ($b$-begin; $e$-end) |

*3.2. Hypothesis*

In this study, we assumed the following conditions.

- There is only one UAV involved;
- The operating time of the UAV is inversely proportional to the load weight; the heavier the load weight of the UAV, the shorter the operating time;
- The service time that the UAV spends on each customer will not be a focus;
- The delivery route is fixed;
- The impact of weather, buildings, or UAV malfunctions on the operating route and speed of the unmanned vehicle will not be considered;
- Once the load weight of the UAV is confirmed in the distribution center, the operating time will be confirmed, and will not increase due to the real-time decreasing load weight during delivery;
- The speed is constant; it will not change due to load requests, number of requests, or operating time;
- The UAV can serve many customers; each customer, however, can only be served once;
- The locations of the UAV and customer are known;
- We assume that the UAV is departing from the logistics center;
- Customer requirements will not change and are known;
- The UAV will return to the logistics center after delivery;
- The load weight carried is inversely proportional to the greatest load weight of the UAV;
- The UAV which departs from the logistics center is completely in-tact, everything runs well, and the UAV comes with a full tank of gas/fully charged battery.

The route planning of this study is derived from the TSPTW and the period of the time during which the node must satisfy the services, or impassability time of the customer. Based on this condition, the route planning must be able to satisfy the request of the customer's period; the time period specified by the customer assigned is a necessary condition (hard time window). In a real case, some customers' acceptable requirements were delayed. Therefore, the target of the logistics service provider is the maximum profit, instead of obtaining the shortest total delivery time for all the customers. The target of this study is to obtain the maximum profit from an unmanned vehicle.

That is the mean of the revenue minus the cost of operating the unmanned vehicle (1). The revenue is the delivery price of the request. The cost includes the variable cost (distance charge minus unmanned vehicle fixed cost), and the penalty cost of missing the time window (penalty cost is divided into delay penalty cost and early arrival penalty cost. The delay penalty coefficient is bigger than the early arrival penalty coefficient). As mentioned above, some customers' acceptable requirements are delayed. Nevertheless, we liberalize the time condition and add a penalty coefficient from the time window in order to deliver on time as often as possible to each node. Finally, we assume the weight of revenue and cost in order to provide different scheduling for the decision-making platform when rush requests occur. (2) After the unmanned vehicle enters one location, it will exit from that location. (3) The unmanned vehicle will avoid transporting within the location. (4) The unmanned vehicle will only serve the same customer once. (5) The time required to complete the total route is the distance divided by the constant speed. (6) The maximum operating time of the unmanned vehicle is such that the operating time of the unmanned vehicle is conversely proportional to the load carried. (7) The required operating time of the unmanned vehicle must be smaller than the maximum operating time. (8) The unmanned vehicle will depart from the current customer to the next customer to avoid looping. (9) There is a penalty cost of the unmanned vehicle missing the time window. There are only three situations: delayed, on time, or early; these cannot occur at the same time, so when the time difference is negative, it will be 0. (10) The rush request condition provides that a rush request is allowed when there is extra capacity for the unmanned vehicle to process a rush request and when the rush request revenue is greater than the rush request cost. After the rush

request is added, request selection can also be processed during the operating procedure of the unmanned vehicle. Certainly, the requirement of the customer in real-time must be less than the capacity of the unmanned vehicle $i$ indicating that the requirement was finished (11). The decision variable is belonging to 0 or 1 (12).

$$Z = \text{Max } w_1 \sum_{j \in C} \sum_{jk \in C} x_{ijk} p_{jk} - w_2 \sum_{j \in C} c_j - w_3 \sum_{j \in C} \sum_{jk \in C} d_{jk} x_{ijk} VC_i - \sum_{i \in D} FC_i \tag{1}$$

s.t.

$$\sum_{i \in D} \sum_{jk \in C} x_{ijk} = \sum_{i \in D} \sum_{jk \in C} x_{ikj} \tag{2}$$

$$x_{ijj} = 0 \tag{3}$$

$$x_{ijk} + x_{ikj} \leq 1 \tag{4}$$

$$T_{(i,d)} = \sum_{j \in C} \sum_{jk \in C} \frac{d_{jk}}{v_i} x_{ijk} \tag{5}$$

$$T_{i,max} = T_{i,0} + \sum_{j \in C} \frac{q_j}{l_i} \tag{6}$$

$$T_{i,d} \leq T_{i,max} \tag{7}$$

$$T_{k,a} \geq T_{j,a} + \left( \frac{d_{jk}}{v_i} \right) - \left( 1 - x_{ijk} \right) \times M \tag{8}$$

$$c_j = \left( T_{j,b} - T_{j,a} \right) c_{j,b} + \left( T_{j,a} - T_{j,a} \right) c_{j,e} \tag{9}$$

$$p_{jk} > d_{jk} VC_i + \sum_{j \in C} c_j \tag{10}$$

$$q_{jk} < Q_i - \sum_{j \in C} q_j \tag{11}$$

$$x_{i,j,k} = \{0,1\} \tag{12}$$

*3.3. Decision of the Heuristic Algorithm*

Since the scheduling and route planning problems have been proven to be NP-hard problems, the study applied a genetic algorithm as the major structure and established a heuristic algorithm to obtain the approximate optimization of the problem. The genetic algorithm uses the solution of the global solution to search, instead of the other algorithm searching from a single solution in the optimal solution. The advantage of searching by the global optimal solution is that it can avoid the situation of falling into the local optimal solution. For this reason, it is a stable method that was used in many studies.

As mentioned above, the decision-making platform is divided into two states, match and real-time. The unmanned vehicle will take in requests in the logistics center and process the request combination of the optimum delivery cost under the known factors of number of requests, delivery conditions, vehicle capacity, delivery region, and the customer's expected time window. Since we already know the customer requirements in the platform, we can process the initial routing, focusing on the known requirements in the match state; the solution computing steps are:

1. Generate random feasible routes.
2. Process reproduction on the selected shorter routes.
3. Pair individuals, chose two nodes randomly, and check the existence of the travel route. A certain subsequent route can be reversed to achieve the result of mutation if nodes conflict.
4. Generate a new plan (offspring Routes + mutation Routes).

However, conflicts may occur during the route planning process. These changes may result in the initial routing not being optimal and the consideration of different situations. Since the temporary requirement occurs in the delivery process in a real-time state, the total time deviation of the entire customer cluster will need to be considered; its purpose is to minimize the deviation. This also reflects that the service time in the new plan should match the service time requirement from the original plan, as much as possible. Thus, if computing a new delivery sequence is required, the steps are:

1. Determine that the vehicle still has space for rush requests if the solution of the match state were satisfied; scheduling should be ended if the vehicle does not have the capacity for new requests.
2. Compute the algorithm to increase revenue and consider customer reception for early or late delivery.
3. Establish corresponding feasible routes for every possible backup schedule.
4. Apply the optimal cost computed in model control of the real-time state as the initial price and calculate every feasible route. If no optimal solution can be computed, recalculate again using the rush request step of the real-time state model evolution.
5. Select the best solution, with minimum operating and delay costs.

The computing process of the heuristic algorithm is shown in Figure 4.

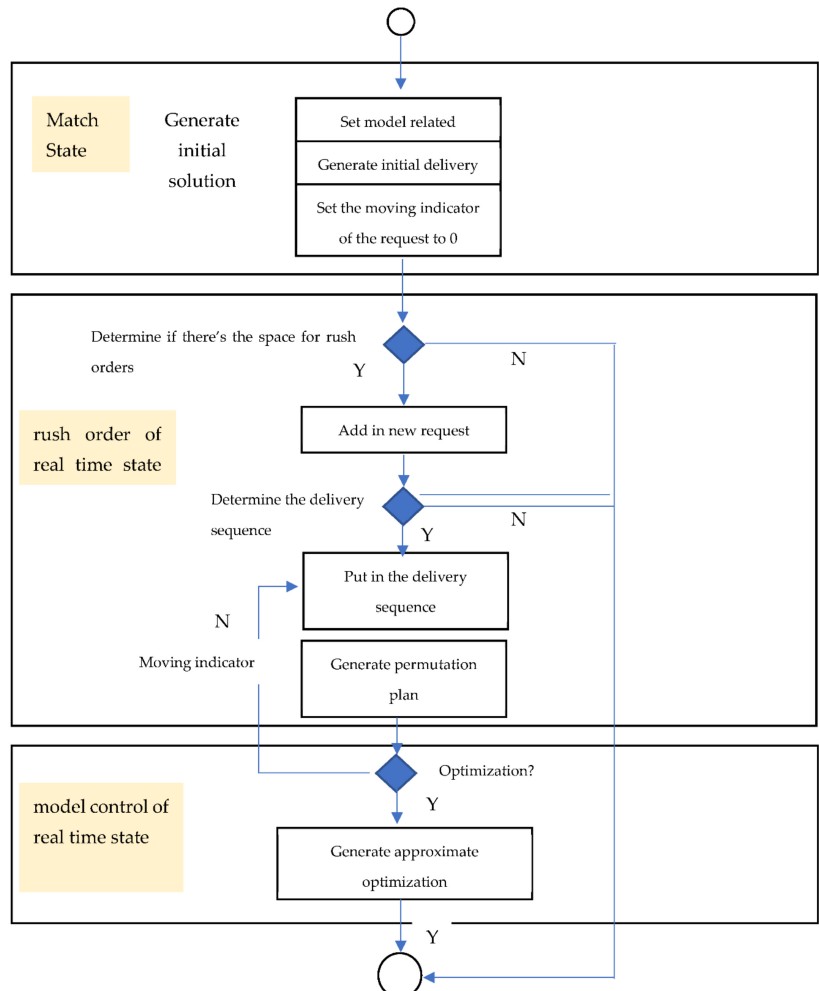

**Figure 4.** The process of the heuristic algorithm in this study.

## 4. Case Study

In Section 3, this study proposed a mathematical model in order to obtain the maximum profit for UAVs, and applied Python 3.6.5 programming to achieve a heuristic

algorithm. In this section, datasets from past studies are cited to be calculated as the result of this study. The result of this study would provide the solution to the decision-making platform of UAVs.

### 4.1. The Result of the Study

The dataset of this study is from https://homepages.dcc.ufmg.br/~rfsilva/tsptw/#literatur (accessed on 12 May 2022). Only the request time window and unit were applied. Both the weight (sourced from the table of random numbers, shown in Appendix A) and related costs (sourced from the price table of Shun-Feng, Wuhan City, China) were processed using one unmanned vehicle. We consider the starting coordinates of the delivery center to be (0, 0), the unmanned vehicle's maximum capacity to be 150, the total flying time to be 400 min, the speed to be 1.5 km/min, the operating time to be inversely proportional to the load weight carried, the inverse ratio to be 0.5, the early arrival penalty coefficient to be RMB 0.1/min, the late arrival penalty coefficient to be RMB 0.2/min, the operating unit cost to be RMB 0.5/km, and the operating fixed cost to be RMB 50. Since the study considers rush request conditions in the model, no similar problems were found in past studies. However, we structured the mathematical model as a linear model in this study. To confirm the accuracy of the algorithm process steps, we used LINGO software to process small-scale testing to compare the experiment results of the heuristic algorithm and mathematical model computed from LINGO in order to inspect the accuracy of the designed algorithm. Under the premise of only selecting 10 requests, the solution computed from LINGO is the same as from the heuristic algorithm; the greater the number of solutions there are to be computed, the more accurate the solution. Therefore, the study considers the result of the heuristic algorithm to be convergent, and that the accuracy of the algorithm can be guaranteed.

Next, we choose one case to verify the result (the related data is shown in Appendix B). In the match state, information of the request is known, and it is calculated by the traditional TSP solution, and the UAVs will execute this solution. The sequence of the route is items 16, 20, 18, 19, and 6 in TSP1 case, and the profit of routing is 102.24. The sequence of routing is (16, 3), (28, 16), (31, 15), (36, 14), and (25, 5). When the new request occurs in real-time, the decision-making platform will add it into the experimental result of the match state, and the algorithm will determine that the new requests will be accepted if the new request can increase revenue more than the scheduling of the match state; otherwise, the request is rejected. Simultaneously, the decision-making platform determines new scheduling in real-time. The simulation result of this study is shown in Figure 5. We find that the new sequence of routing is (5, 4), (28, 16), (31, 15), (36, 14), (25, 5), and (16, 3). Comparing the two results between the match and real-time state, the decision-making platform would select the request near the service center.

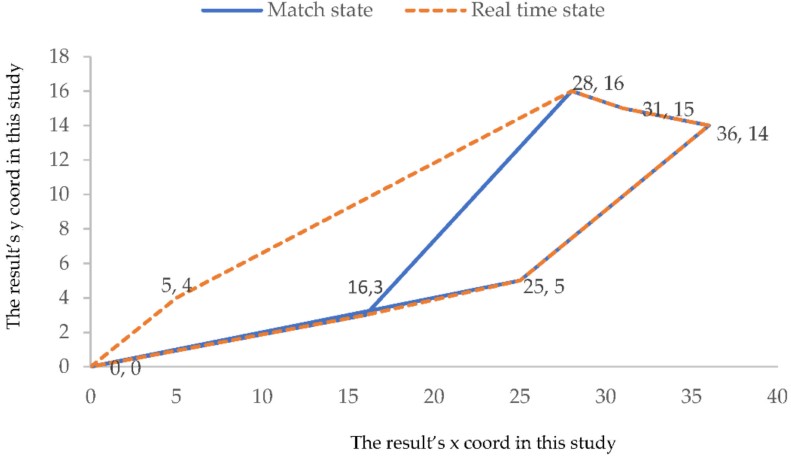

**Figure 5.** Difference result between match state and real-time state.

### 4.2. Sensitivity Analysis

The study model is different from the traditional vehicle routing problem. The total delay penalty and the early penalty amounts are introduced to the study by taking customer dissatisfaction and the total missed time window into account. To understand the impact of the total route distance, time window penalty amount, and total company profit, the study conducted a sensitivity analysis on different weighted parameters. The study found that the premise of the weighted penalty of missing the expected time window is confirmed; the profit will decrease when the minimum route weight increases. At the same time, the total missed time window will decrease. However, the decreasing level of profit is obviously much greater than the missed time window. When the logistics service providers require the shortest route, they will only receive new requests that are near the logistic center, which further limits the request selection amount and results in a decrease in profit. Furthermore, the unmanned vehicle will not often miss the customer's expected time window due to the decreasing requests; therefore, missed time windows will also decrease (shown in Figure 6).

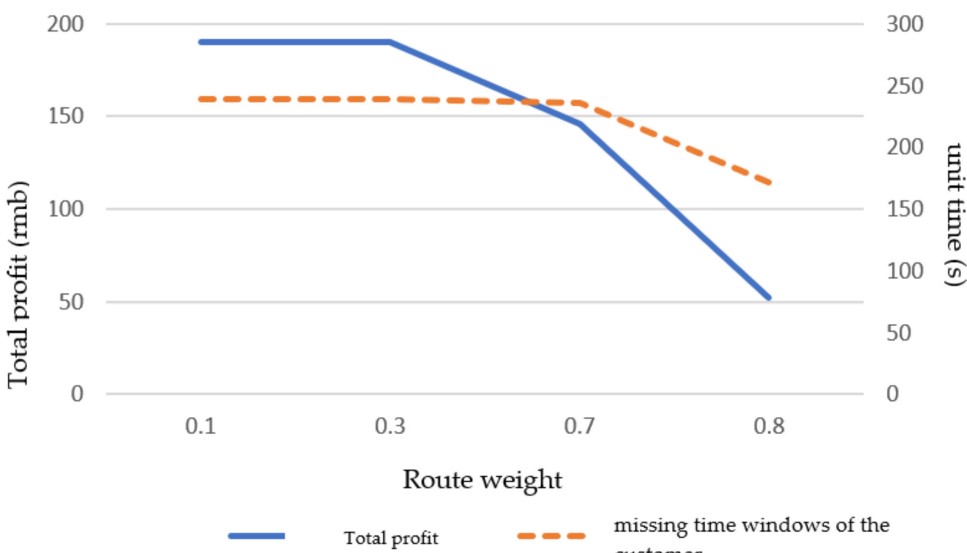

**Figure 6.** The result of correlations between the shortest route weight, the profit, and missed time windows.

Figure 7 shows the impact of changing the weight of missing time windows on the profit and total route distance under the premise of a fixed shortest route weight. Noticeably, we can see the revenue decrease along with the increase in the weight of missed time windows. The reason for this is that under the premise of the company requiring fewer missed time windows, the number of requests that can be accepted is reduced, thus the profit decrease. There is no fixed changing trend for the shortest route distance; it continuously decreases with the fluctuations, along with the increasing weight of missed time windows. The reason for this is that the time window of the request around the logistics center does not have a fixed changing trend; the shorter the traveling distance, however, the less likely it will miss the time window; thus, the shortest route distance will constantly fluctuate.

Figure 8 shows the impact of changing the weight of missed time windows on the profit and total route distance under the premise of fixed profit. Noticeably, we can see that the profit and total route distance increase along with the increased missed time window weight. The reason for this is that we can only try to reduce the total route distance and maximize the profit, while the weight of missed time windows constantly increases. In the end, once the profit and total route distance are stable, the approximate optimal solution has been found (maximum revenue with the fewest missed time windows).

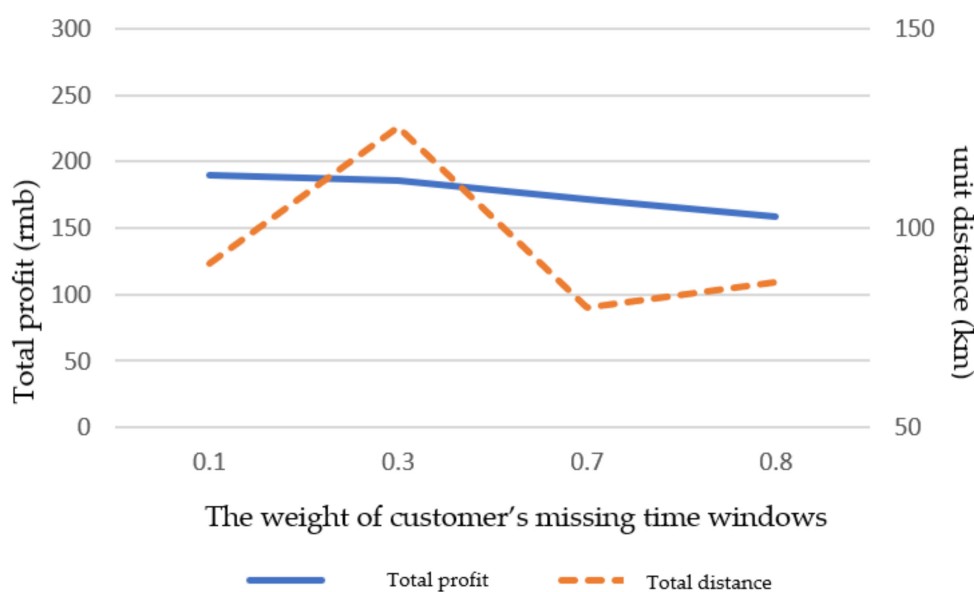

**Figure 7.** The result of the relationship between the weight of missed time windows and the profit and total route distance under a fixed shortest route weight.

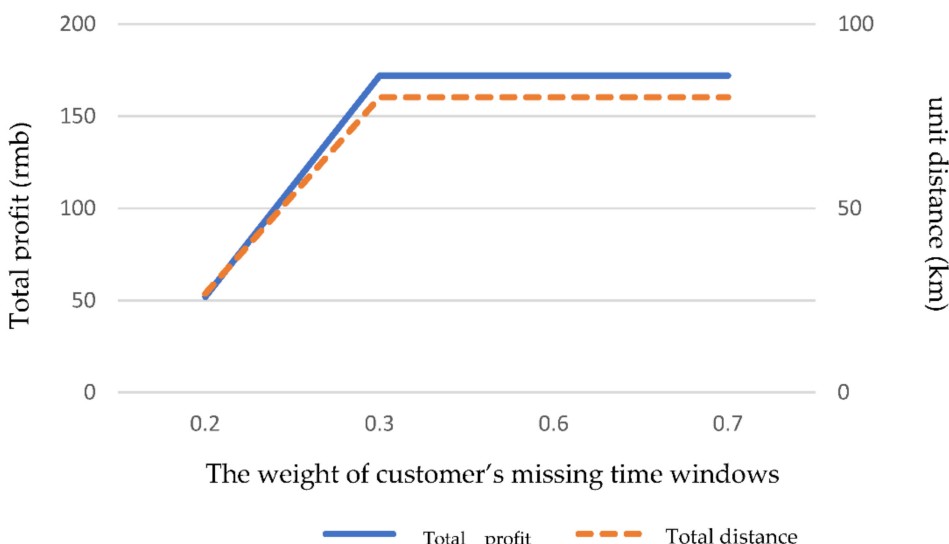

**Figure 8.** The result of the relationship between the weight of missed time windows and the profit and total route distance under the logistics service provider's profit.

Traditionally, the process of receiving a request and scheduling are considered two separate functions on the decision-making platform of the logistics provider. The logistics service provider proposed the route planning after confirming the request of the customer. Technological improvement is a factor of transformations in urban delivery. Specifically, UAV implementation by logistics service providers will reduce the limitations of urban delivery. The logistics service provider can receive the new request in order to obtain revenue, rather than considering limitations in real-time. However, if a new request is added to the routing, it may not be a good solution, because the customer will be dissatisfied if the expected time window fails to meet their expectations.

Thus, this study has proposed an equivalent mathematical model that focuses on the new route planning problem. We liberalize the time condition and add a penalty coefficient to the time window in order to deliver on time as often as possible to each node and

maintain customer satisfaction. With the results of this study, we can explain the situation of the logistics service provider.

1.   The logistics service providers decided to increase the weight of customer satisfaction to achieve the least missed time windows. Because the time window of the request around the logistics center does not have a fixed changing trend, the shortest route distance will constantly fluctuate.
2.   While increasing the weight of the shortest route can decrease the missed time windows, it will lower the profit at the same time. The logistics service providers will only receive a new request that is near the logistic center.
3.   The profit and total route distance increase along with the increased missed time window weight. This is mean that the logistics service provider can receive the new request when the customer can accept a delay in their request.

Finally, we changed the maximum load capacity, the total flight time, the speed, and the carrying capacity of the UAV to explore the relationship with profit. The study found that most of the changes in UAVs setting had no influence, but changing the load of the UAV can be an influencing factor in the sales and total profit of the UAV, because logistics operators can receive more requests, which are related to customers. We can discuss that the customer is a key factor in the logistics decision-making platform, not the provider investment.

## 5. Conclusions and Recommendation

In this study, the driving force for UAV technology application in the logistics industry is summarized into three points: the rapidly increasing delivery demand, increasing labor costs, and complicated service scenarios. Delivery companies have started facing high labor costs and difficult terminal delivery; how to deliver requests to customers faster has become the greatest challenge for the provider. UAVs can fulfill the requirements for receiving/delivering services to office buildings, private parks, campuses, and other request-concentrated areas in cities. They can significantly improve delivery performance with their unique flexibility and convenience.

Through allocation, diverse types of UAVs will apply the related program algorithms to coordinate the operation, which helps the logistics service provider to achieve as yet unseen high performance, speed, and accuracy. The development of the logistics industry will seamlessly connect suppliers with consumers in the future. It will accomplish the personalized experience of accurate request matching, increasing product value through logistics and rapid product circulation. According to the data provided by Chinese e-commence, applying UAV delivery in remote areas can increase performance by 50~60%, and by adding in rush requests, it can fully use the remaining resources to bring greater revenue for companies.

The study applied the penalty factors to the original TSP for promoting customer satisfaction. There are two penalty factors: delay, and early arrival. This offers the maximum profit scheduling and determines whether a rush request can be accepted by combining the unmanned vehicle's limited load weight capacity, and the fact that operating time is inversely proportional to the load weight carried. The rush request will be accepted if the revenue from running the UAV can increase. Otherwise, the rush request will be rejected. This will further optimize the unmanned vehicle's schedule. The study considers the relationship between customer satisfaction, load weight, and operating time. Customer loyalty will increase based on the decision-making process, if customer satisfaction is considered when addressing the shortest distance. The model algorithm established by the study can also be regarded as a useful decision-making tool.

The study is the application of TSP in UAV cases. It has included the UAV's limited load weight or capacity, and that its operating time is inversely proportional to the load weight carried. The study designs the environmental condition that does not require all requests to be delivered to the hub or the logistics center, but to be processed directly by the calculation of the schedule for the UAV, considering the maximum profit. Furthermore, it

can also improve the traditional VRP, and multiple unmanned vehicles can work together to maximize total profit. In addition, the study only includes one rush request. It can be expanded to contain multiple rush requests in the future, which can retrieve requests in various locations first, then process delivery afterward. After the rush request is accepted, the remaining resources can be more effectively allocated to save more resources for companies, further obtain more profit, complete more requests, and guarantee the market share.

The delivery fee standard should include weight and volume. In addition to load weight carried, a UAV also has volume limitations. This study has only discussed the limit of load weight; volume cost and the load volume of a UAV can also be included in the future to create a more realistic scenario and further provide the industry with more references and recommendations. Along with people's constantly improving living standards, the logistics service provider will need to pay more attention to customer satisfaction; therefore, we can also adjust the corresponding weights of penalty factors. In the event of an unpleasant experience because of the missed time windows, customer compensation can be offered, which can help to avoid dissatisfaction. Future work will study the acceptable amount of compensation for customers and the applicable situations for compensation.

**Author Contributions:** Conceptualization, I.-C.L. and S.-H.C.; methodology, I.-C.L. and S.-H.C.; software, I.-C.L.; validation, I.-C.L. and T.-H.L.; formal analysis, I.-C.L.; investigation, I.-C.L.; data curation, I.-C.L.; writing—original draft preparation, I.-C.L.; writing—review and editing, I.-C.L.; visualization, S.-H.C.; supervision, T.-H.L.; project administration, T.-H.L. All authors have read and agreed to the published version of the manuscript.

**Funding:** This research received no external funding.

**Conflicts of Interest:** The authors declare no conflict of interest.

## Appendix A. Random Table of the Weight of the Demand

In this study, we use the program software (Python 3.6.5.) to generate the random number table in order to decide the weight of the demand. Every set is a $5 \times 5$ matrix.

| | | | | |
|---|---|---|---|---|
| 03 47 43 73 86 | 36 96 47 36 61 | 46 98 63 71 62 | 33 26 16 80 45 | 60 11 14 10 95 |
| 97 74 24 67 62 | 42 81 14 57 20 | 42 53 32 37 32 | 27 07 36 07 51 | 24 51 79 89 73 |
| 16 76 62 27 66 | 56 50 26 71 07 | 32 90 79 78 53 | 13 55 38 58 59 | 88 97 54 14 10 |
| 12 56 85 99 26 | 96 96 68 27 31 | 05 03 72 93 15 | 57 12 10 14 21 | 88 26 49 81 76 |
| 55 59 56 35 64 | 38 51 82 46 22 | 31 62 43 09 90 | 06 18 44 32 53 | 23 83 01 30 30 |
| 16 22 77 94 39 | 49 54 43 54 82 | 17 37 93 23 78 | 87 35 20 96 43 | 84 26 34 91 64 |
| 84 42 17 53 31 | 57 24 55 06 88 | 77 04 74 17 67 | 21 76 33 50 25 | 83 92 12 06 76 |
| 63 01 63 78 59 | 16 95 55 67 19 | 98 10 50 71 75 | 12 86 73 58 07 | 44 39 52 38 79 |
| 33 21 12 34 29 | 78 64 56 07 82 | 52 42 07 44 38 | 15 51 00 13 42 | 99 66 02 79 54 |
| 57 60 86 32 44 | 09 47 27 96 54 | 49 17 46 09 62 | 90 52 84 77 27 | 08 02 73 43 28 |
| 18 18 07 92 45 | 44 17 16 58 09 | 79 83 86 19 62 | 06 76 50 03 10 | 55 23 64 05 05 |
| 26 62 38 97 75 | 84 16 07 44 99 | 83 11 46 32 24 | 20 14 85 88 45 | 10 93 72 88 71 |
| 23 42 40 64 74 | 82 97 77 77 81 | 07 45 32 14 08 | 32 98 94 07 72 | 93 85 79 10 75 |
| 52 36 28 19 95 | 50 92 26 11 97 | 00 56 76 31 38 | 80 22 02 53 53 | 86 60 42 04 53 |
| 37 85 94 35 12 | 83 39 50 08 30 | 42 34 07 96 88 | 54 42 06 87 98 | 35 85 29 48 39 |
| 70 29 !7 12 13 | 40 33 20 38 26 | 13 89 51 03 74 | 17 76 37 13 04 | 07 74 21 19 30 |
| 56 62 18 37 35 | 96 83 50 87 75 | 97 12 55 93 47 | 70 33 24 03 54 | 97 77 46 44 80 |
| 99 49 57 22 77 | 88 42 95 45 72 | 16 64 36 16 00 | 04 43 18 66 79 | 94 77 21 21 90 |
| 16 08 15 04 72 | 33 27 14 34 09 | 45 59 34 68 49 | 12 72 07 31 45 | 99 27 72 95 14 |
| 31 16 93 32 43 | 50 27 89 87 19 | 20 15 37 00 49 | 52 85 66 60 44 | 38 68 88 11 80 |
| 68 34 30 13 70 | 55 74 30 77 40 | 44 22 78 84 26 | 04 33 46 09 52 | 68 07 97 06 57 |
| 74 57 25 65 76 | 59 29 97 68 60 | 71 91 38 67 54 | 13 58 18 24 76 | 15 54 55 95 52 |
| 27 42 37 86 53 | 48 55 90 65 72 | 96 57 69 36 10 | 96 46 92 42 45 | 97 60 49 04 91 |
| 00 39 68 29 61 | 66 37 32 20 30 | 77 84 57 03 29 | 10 15 65 04 26 | 11 04 96 67 24 |
| 29 94 98 94 24 | 68 49 69 10 82 | 53 75 91 93 30 | 34 25 20 57 27 | 40 48 73 51 92 |
| 16 90 82 66 59 | 83 62 64 11 12 | 67 19 00 71 74 | 60 47 21 29 68 | 02 02 37 03 31 |
| 11 27 94 75 06 | 06 09 19 74 66 | 02 94 37 34 02 | 76 70 90 30 86 | 38 45 94 30 38 |
| 35 24 10 16 20 | 33 32 51 26 38 | 79 78 45 04 91 | 16 92 53 56 16 | 02 75 50 95 98 |
| 38 23 16 86 38 | 42 38 97 01 50 | 87 75 66 81 41 | 10 01 74 91 62 | 48 51 84 08 32 |
| 31 96 25 91 47 | 96 44 33 49 13 | 34 86 82 53 91 | 00 52 43 48 85 | 27 55 26 89 62 |

## Appendix B. The Example (Customer Demand) of This Study

The data set is from https://homepages.dcc.ufmg.br/~rfsilva/tsptw/#literatur (accessed on 12 May 2022). We used the datasets proposed by Dumas et al., and chose TSP1 recorded in n20w20 sets for this study.

| !! n20w20.001 | | | | | | |
|---|---|---|---|---|---|---|
| CUST NO. | XCOORD | YCOORD | DEMAND | READY TIME | DUE DATE | SERVICE TIME |
| 1 | 16.00 | 23.00 | 0.00 | 0.00 | 408.00 | 0.00 |
| 2 | 22.00 | 4.00 | 0.00 | 62.00 | 68.00 | 0.00 |
| 3 | 12.00 | 6.00 | 0.00 | 181.00 | 205.00 | 0.00 |
| 4 | 47.00 | 38.00 | 0.00 | 306.00 | 324.00 | 0.00 |
| 5 | 11.00 | 29.00 | 0.00 | 214.00 | 217.00 | 0.00 |
| 6 | 25.00 | 5.00 | 0.00 | 51.00 | 61.00 | 0.00 |
| 7 | 22.00 | 31.00 | 0.00 | 102.00 | 129.00 | 0.00 |
| 8 | 0.00 | 16.00 | 0.00 | 175.00 | 186.00 | 0.00 |
| 9 | 37.00 | 3.00 | 0.00 | 250.00 | 263.00 | 0.00 |
| 10 | 31.00 | 19.00 | 0.00 | 3.00 | 23.00 | 0.00 |
| 11 | 38.00 | 12.00 | 0.00 | 21.00 | 49.00 | 0.00 |
| 12 | 36.00 | 1.00 | 0.00 | 79.00 | 90.00 | 0.00 |
| 13 | 38.00 | 14.00 | 0.00 | 78.00 | 96.00 | 0.00 |
| 14 | 4.00 | 50.00 | 0.00 | 140.00 | 154.00 | 0.00 |
| 15 | 5.00 | 4.00 | 0.00 | 354.00 | 386.00 | 0.00 |
| 16 | 16.00 | 3.00 | 0.00 | 42.00 | 63.00 | 0.00 |
| 17 | 25.00 | 25.00 | 0.00 | 2.00 | 13.00 | 0.00 |
| 18 | 31.00 | 15.00 | 0.00 | 24.00 | 42.00 | 0.00 |
| 19 | 36.00 | 14.00 | 0.00 | 20.00 | 33.00 | 0.00 |
| 20 | 28.00 | 16.00 | 0.00 | 9.00 | 21.00 | 0.00 |
| 21 | 20.00 | 35.00 | 0.00 | 275.00 | 300.00 | 0.00 |
| 999 | 0.00 | 0.00 | 0.00 | 0.00 | 0.00 | 0.00 |

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
