# Peer review of "A Decision System for Routing Problems and Rescheduling Issues Using Unmanned Aerial Vehicles"

_applsci, doi:10.3390/app12126140_

Round 1
Reviewer 1 Report
The paper is about a new vehicle routing problem model for UAVs, considering the customer satisfaction in the model. This research can lead to an important development in city logistics regarding drone deliveries and can help several related projects and research with this new VRP solution.
The paper presents interesting and important new research results, but there are several problems with its structure, expressions, and language; several minor corrections are required, several expressions should be changed, some minor changes are required in its structure, and its language must be thoroughly checked.
Regarding the structure and the content of the paper, I have the following comments:
- please, make the target of the examined model clear in the abstract (for example, customer satisfaction is considered as well), and describe in the abstract, that exactly what is new in your model
- "More than one hundred cities have started unmanned vehicle trial operations." (in lines 33 and 34); regarding this, I am missing a reference, or if it is the result of own analysis, there should be a table with the data in the appendix section, as these projects are important inputs for such research
- in the first paragraph of the introduction, the capacity and the range limitations of drones in city logistics should be addressed as well, as these are very important factors for urban deliveries, and these are actually the biggest problems regarding the use of drones in an urban environment
- the second paragraph of the introduction is extremely long; it should be restructured by the different topics within VRP; in its actual form, it is really difficult to follow it (for example, some parts could be presented in tables instead of longer text)
- in Figure 2, the "Delivery order generated" box has a different font type; please, use the same font type on the whole figure
- in the first paragraph of section 2.2., the meaning of soft and hard time windows should be explained
- I don't understand the name of Table 1; it should be changed to a clear title, like "The variables used in the developed model."
- in Table 1, the meaning of "M" ("A big constant") should be explained; for what is this constant value used in the model?
- in Table 1, i, j, k, etc., should be in quotes; for example, Fixed travel cost of the unmanned vehicle "i"
- in Table 1, the font types of the symbols are different; please, change all of them to the same font type
- in Table 1, at the symbols of Unit distance from j to k, Time arrived at customer j.k of unmanned vehicle (a-arrive), Total operating time of unmanned vehicle i, Time window lower limit of j.k and Time window upper limit of j.k (b-begin; e-end), there are some not necessary underlines, and the commas are used incorrectly, please, change it to make It more understandable
- in line 210, the first equation (Z = …) should be numbered as well
- the font sizes of the symbols in the equations are not consistent; please, correct them; at the same time, their numbering is not exactly above each other; please, correct that too
- the reference for Figure 3 is one page earlier than the figure itself; the figure should come directly after its description in the text
- at the start of section 4 (in lines 287-288), the section should be shortly presented before starting subsection 4.1
- section 4 should be called "Case study" instead of "Case Analysis"
- in line 290, "https://homepages.dcc.ufmg.br/~rfsilva/tsptw/#literatur" should be added with a numbered reference instead, and its data should be shown in the References section
- in line 302, the meaning of "LINGO" should be presented
- in Figure 4, the names and the dimensions of the axes are missing; please, add them
- in Table 2, the dimensions are missing; please, add them
- based on my opinion, Figures 5, 6, and 7 are not necessary; the information on them should be only described in the body text, and the description from line 320 to line 341 should be rewritten; it is really difficult to follow it in its actual structure, the lists should be reorganized
- in Figure 8, the names and the dimensions of the axes are missing; please, add them
- in Figures 9, 10, and 11, the dimensions of the axes are missing, and the name of the y-axes is basically out of the figure; please add it to the chart itself, and use only one font type on the figure (this is true for all the other figures as well)
- in the abstract, it was stated that "UAV investment has no significant impact on total profit and customer satisfaction", in the paper, this became not clear to me; please, write about this point more and make this conclusion clear, or remove it from the abstract
- in Appendix A, the data should be added in better quality (on a better-quality figure, or it would be even better to add it as a table)
- Appendix B should be added as a table with exact, well understandable column names
I have some comments regarding the expressions used in the paper:
- instead of "taking into account", "considering" would be the better expression (for example, in lines 18 and 19)
- "last kilometer" of the delivery (for example, in lines 29 and 37) should be called "last mile" instead; that is the most common expression for this
- in line 42, instead of "boring", "usual" or "basic" would be the appropriate word, and instead of "dangerous", "special" would be definitely better
- instead of "logistical efficiency" (for example, in line 105), please, use "logistics efficiency."
- instead of "E-commerce", "e-commerce" is the correct expression without a capital letter; please, change it everywhere in the document (for example, in line 113)
- instead of "firstly" (for example, in line 139), "first" should be used everywhere
- instead of "minimal" (for example, in line 244), "minimum" should be used everywhere in the paper
- instead of "vehicle route problem" (for example, in line 347), "vehicle routing problem" should be used
Regarding the language and the spelling mistakes, I have the following comments:
- line 44: "difficult" instead of "difficulties", please, correct it
- in the sentence starting in line 56 is incorrect, the verb is missing, please, correct the sentence
- the sentence finishing in lines 65, and 66 is "finished twice", please, remove the "." before "(Li et al., 2007)"
- in line 95, the commas before and after "air traffic control" are not correct; some special characters are used instead; please, change it to commas
- in lines 113 and 144, "demands" should be there instead of "demand"; please correct it
- in the sentence starting in line 120, from the "Different from the 120 commonly seen delivery model" part, the verb is missing; please, correct it
- in Figure 2, instead of "Take new order", it should be "Taking new order"; please, change it on the figure
- in line 155, "we assumed the condition" should be changed to "we assumed the following conditions."
- in line 173, "We assume the unmanned vehicle departs from the logistics center" should be changed to "We assume that the unmanned vehicle is departing from the logistics center"
- in line 186, "development model" should be changed to "developed model"
- in line 193, ")" is missing; please, add it
- in equation (5), there is a not necessary underline; please, correct that
- in line 205, there is (1.10) instead of (10); please, correct it
- in line 209, a space is missing before (11); please, correct it
- in line 295, a space is missing before "km/min"; please correct it
- in Table 2, in column 4, the positioning of the two rows is different; please, make them the same
- in line 368, a comma is missing before "we"; please, correct it
- in line 383, a comma is missing before "we"; please, correct it
Author Response
First, I think I must thank your comments, these comments let the paper easy to read for the reader. Next, I responded to the comments sequentially.

Reviewer 2 Report
The paper focuses on the potential usage of Unmanned Aerial Vehicles for urban logistic mobility. The routing and scheduling problems of the Unmanned Aerial Vehicles are offered to solve with a genetic algorithm. The paper is readable, however, there are some possessive pronouns problems and grammar errors, which are needed to be fixed. The paper organization is not fine, the paper must be re-organized totally (e.g., such as adding a Methodology section!!, also some current section titles are potential subtitles for this study). The text format in the figures could be more professional. Also, the visualization quality of some figures is poor (e.g., figures 4, 8), also figures 6,7,8 are presented as a copy-paste, I suggest sharing them in the table format. In general, I suggest also more alive coloring. The reasoning for the need for the UAV in urban mobility must be more elaborated by harmonizing the current situation of the urban mobility system (e.g., https://doi.org/10.3390/su131810158) in the Introduction. The potential contribution of the study could be expressed better in the last paragraph of the Introduction. The methodological definition and reasoning of the method selection are missing. What is the genetic algorithm and why choose it over the current other algorithms that could also be used?? As well, some basic technical terms regarding the used methodology should be explained for the multi-disciplinary readers. The assessment of the results is quite poor; therefore, I could not evaluate this section! The article definitely needs a major revision, thereafter I will reconsider my decision.
Author Response

(The authors gave the same response as above.)

Reviewer 3 Report
The paper covers the scope of the Journal. The Authors present a contemporary problem, i.e. the TSP solved with the application of the proposed mathematical model assuming the usage of unmanned vehicles with their constraints, such as limited load weight capacity and operating time. A conferred literature review is relevant.
My concern is a quality of information presented, since some parts of the paper are ambiguous or not clear, e.g.:
- The Authors define the TSP problem in line 59 and they show the relations between TSP and VRP. However, the definition of the VRP is not provided.
- The research gap is not highlighted and it should be clarified.
- The business model of logistics industry delivery (see figure 1) is not well presented and described, i.e. information and product flows are not distinguished. What is the role of new requests? It is not clear who is submitting those new requests. What is the stopping condition? What is the difference between “Client” and “Client”?
- The Authors mention about the “information platform” (see line 133) – is this expression a synonym of “decision making platform” presented in figure 1?
- The description of the mathematical model is not precise. The Authors do not state that there are described constraints; there are only the numbers (1-11). Moreover, not all information presented in section 2.2 is coherent with the mathematical formulas, e.g. Pjkv. Pj.
Objective function
a) The objective function includes parameters, such as W1, W2 and W3. What is their meaning?
b) Is the objective function calculated for each unmanned vehicle i? If yes, it should be denoted in the objective function.
c) The second component of the objective function, i.e. penalty expense of the unmanned vehicle not arriving within the time window, is not related to the decision variable. In that sense it does not matter whether the delivery will be made within the time window or not, because the penalty is always calculated. I would appreciate if the Authors could comment on that part of the mathematical model.
d) The third component of the objective function represents the distance measure, i.e. there are two parameters related to the distance. They are as follows:
- dijunit distance from j to k,
- VC travel unit distance of unmanned vehicle i.
Since the mathematical formula of this component of the mathematical model relates to distances, I have two questions:
- Why are those two distances multiplied?
- What is the unit of the objective function if there is calculated the difference between costs (component one, two and four) and distance (component three)?
Constraints
a) Most of the constraints are not well described. Some examples of my questions and comments are presented below:
b) Constraint (1):
What is “D”?
c) Constraint (3):
Left and right side of the equation is the same.
d) Constraint (5):
Total operating time of unmanned vehicle is the function of distance and speed, but the last two parameters should be divided (not multiplied).
e) Constraint (6):
What is Ti,0?
What is li– the load carried by the unmanned vehicle? Since the Authors define it as “the load of the unmanned vehicle is inversely proportional to the fly time”, is it expressed by the mathematical formula?
What is qj? The Authors define it as “customer j’s requirement”. Is it the amount of the ordered load or the time window of delivery or the other requirement?
f) Constraint (8):
What is Ti,u?
g) Constraint (9):
The condition of the situation when Tj,a> Tj,bis not mathematically expressed.
The second component of the sum is always zero.
h) Constraint (10):
The component of the order price, i.e. djk*VC, does not include the cost factor.
What is qnow?
- What is "step B" mentioned in lines 242 and 243?
- Information presented in figure 4 is not clear. What is the meaning of the blue and orange colors? What values are presented on the ordinate axis?
- Information in table 2 is not well described. What is “item 15, 19, 17, 18, 05” in column “The result of scheduling”?
- Information presented in lines 320 – 341 is ambiguous. It is hard to find the beginning of the figures.
- Information presented in figure 8, Appendix A and Appendix B should be precisely described.
The paper is well written. However, English changes, including punctuation, spelling and grammar are required as well as some editing corrections.
In my opinion, a profound revision of the text should be carried out, including the mathematical model and the description of the computational results.
Author Response

(The authors gave the same response as above.)

Round 2
Reviewer 2 Report
The authors show an effort to revise the manuscript, but still the English level could be better to increase understanding, please add also DOIs of the cited literature before publication, and reference 9 should be cited earlier in the line 43 (Technological improvement...) not in line 66.
Author Response

(The authors gave the same response as above.)

Reviewer 3 Report
Title: A Decision System for Routing Problems and Rescheduling in Unmanned Aerial Vehicles
REVISION
I would like to thank the Authors for the revised version of the paper and their responses to my comments.
My recommendation is: accept after minor revision.
Moderate English changes, including punctuation, spelling and grammar are required. The sentences in lines: 17-18, 25-26, 41-42, 44-64, 148-150, 486-488, 531-532, 539-540, 728, 972, 962-963 (what are the “items”?), 1287-1289, 1303-1304 should be verified because of their ambiguity. It seems that they are not finished, something is missing or they are not clear enough.
I also suggest the following changes:
- Heading of table 2 should be verified.
- Some decisions Y/N in Figure 3 should be added or deleted.
- Values of parameters presented in lines 944-949 should also include symbols of the parameters presented in the mathematical model.
- Figure 4 should be precisely described.
I have the following comments on the mathematical model:
1. Mathematical model includes the indices j and k, which have different meanings. They are used with the decision variable, and they are defined as the parameter, as well. The former means the customers from j to k served by the unmanned vehicle, while the latter means the customer in the real-time state. I would appreciate if the Authors could comment on that.
2. What is the measure of the parameter “cj,b – an unmanned vehicle arriving before the start of customer j’s time window”?
3. The objective function (1):
- What are: w1, w2 and w3?
– Why is the delivery price maximized? I assume that the Z should be maximized.
4. The constraints:
- Is the component qj/lj of the constraint (6) measured in time units?
- What is Ti,u in the mathematical formula (8)?
- What is the unit of the component (1-xijk)xM in the time constraint (8)?
- What is the cj,e in the constraint (9)?
- What is Cj (capital letter C) in the constraint (10)?
My suggestion is to add the units to the description of the parameters.
Author Response

(The authors gave the same response as above.)
